# Comparison of the Utility of RP-TLC Technique and Different Computational Methods to Assess the Lipophilicity of Selected Antiparasitic, Antihypertensive, and Anti-inflammatory Drugs

**DOI:** 10.3390/molecules24173187

**Published:** 2019-09-02

**Authors:** Alina Pyka-Pająk, Wioletta Parys, Małgorzata Dołowy

**Affiliations:** Department of Analytical Chemistry, School of Pharmacy with the Division of Laboratory Medicine in Sosnowiec, Medical University of Silesia in Katowice, PL-4 Jagiellońska Street, 41-200 Sosnowiec, Poland

**Keywords:** lipophilicity, antiparasitic drugs, antihypertensive drugs, anti-inflammatory drugs, Soczewiński–Wachtmeister’s equation, Ościk’s equation

## Abstract

The aim of this study was to assess the lipophilicity of selected antiparasitic, antihypertensive and non-steroidal anti-inflammatory drugs (NSAIDs) by means of reversed phase–thin layer chromatography (RP-TLC) as well by using Soczewiński–Wachtmeister’s and J. Ościk’s equations. The lipophilicity parameters of all examined compounds obtained under various chromatographic systems (i.e., methanol-water and acetone-water, respectively) and those determined on the basis of Soczewiński-Wachtmeister’s and Ościk’s equations (i.e., R_MWS_ and R_MWO_) were compared with the theoretical ones (e.g., AlogPs, AClogP, milogP, AlogP, MlogP, XlogP2, XlogP3) and the experimental value of the partition coefficient (logP_exp_). It was found that the R_MWS_ parameter may be a good alternative tool in describing the lipophilic nature of biologically active compounds with a high and low lipophilicity (i.e., antihypertensive and antiparasitic drugs). Meanwhile, the R_MWO_ was more suitable for compounds with a medium lipophilicity (i.e., non-steroidal anti-inflammatory drugs). The chromatographic parameter φ_0(a)_ can be helpful for the prediction of partition coefficients, i.e., AClogP, XlogP3, as well as logP_exp_ of examined compounds.

## 1. Introduction

Lipophilicity is an important parameter widely used in medicinal chemistry in the design of new drug candidates. This physicochemical property provides the most important information about a proper biomolecule, including its binding to a suitable receptor and ADMET (Absorption, Distribution, Metabolism, Excretion and Toxicity) properties [1]. What is more, in combination with steric and electron factors, it is a powerful tool in the development of novel drugs and their analogues with a greater effectiveness by means of SAR (Structure-Activity Relationship) and QSAR (Quantitative Structure-Activity Relationship) studies, respectively [2,3,4]. Different theoretical as well as experimental methods for determining the lipophilicity are distinguished [5,6]. Among them, the most frequently used are those based on a simple extraction in the *n*-octanol-water system such as the classical *shake-flask* method [5,6]. However, due to limitations, this procedure is frequently being replaced by chromatographic techniques.

The current literature review indicates that reversed-phase liquid chromatography (RP-LC) is an alternative method for the determination of the lipophilicity of various biocompounds by modeling the distribution of a proper compound between lipid layers and biological membranes of cells [5]. The first well known kind of liquid chromatography that was widely used to establish the lipophilicity of various chemical compounds was thin-layer chromatography (TLC). This technique performed in reversed phase system (RP-TLC) has many advantages over the *shake-flask* method. Moreover, a higher reproducibility of the obtained lipophilicity results can be observed [3,5].

In addition to experimental methods, in order to determine the *n*-octanol-water partition coefficient (logP), different theoretical methods (computer programs) allowing the initial assessment of the logP value can also be applied [3]. Many various lipophilicity descriptors and methods have been well described and proposed for the evaluation of the lipophilic properties of different compounds in review articles from recent years [7,8,9]. Currently, the chromatographic approach is used the most [10,11,12,13,14,15,16,17,18]. 

The knowledge concerning the hydrophilic-hydrophobic properties of studied antiparasitic drugs in the current literature is spare and contradictory. For example, Biagi [19] has reported that tididazole and metronidazole are hydrophilic compounds. Másson et al. [20] has shown that metronidazole is a very hydrophilic compound. However, according to Hoffmann et al. [21] and Jokipii et al. [22], metronidazole is a lipophilic compound that penetrates well into most tissues. Next, Guerra et al. [23] studied the relationship between the values of the chromatographic parameter R_M_ of antiparasitic drugs and their logP values. The RP-TLC technique was used to determine the R_M_ values of metronidazole, ornidazole and tinidazole, but no secnidazole was studied in this case. The lipophilicity parameter (R_MW_) estimated by Soczewiński-Wachtmeister’s equation was 0.08 for metronidazole, 0.33 for ornidazole, and 0.35 for tinidazole, respectively. It was also noted, that there are significant differences between the theoretically calculated and experimentally determined logP values of nitroimidazoles, similarly like in the study by Lin et al. [24]. 

Moreover, Guerra et al. [23] studied the R_M_ values of metronidazole, ornidazole, and tinidazole in the chromatographic systems, where the stationary phase was squalene, undecane and liquid paraffin. Additionally, the described R_M_ parameter was measured on unimpregnated layers of silica gel. The lipophilicity parameter (R_MW_) obtained by using Soczewiński-Wachtmeister’s equation indicated that the addition of silicone oil to the stationary phase can modify the partitioning of the examined compound between both, i.e., the stationary phase and mobile phase, and thus their chromatographic parameter of lipophilicity. 

The lipophilicity of selected antiparasitic drugs was also investigated by Guerra et al. using the RP-HPLC technique [25]. The measured logk’ value for metronidazole was –0.067, for ornidazole it was 0.199 and for tinidazole it was –0.125, but secnidazole was not studied. The values of this chromatographic parameter correlated better with the logP values determined in the *n*-octanol-water system than the previously determined R_M_ parameters of the tested compounds [23]. 

Next, Sârbu et al. [26] investigated the lipophilicity of selected non-steroidal anti-inflammatory drugs (NSAIDs), e.g., indomethacin, ketoprofen and phenylbutazone, by using the RP-TLC technique, but did not study mefenamic acid, nabumetone, carprofen and flurbiprofen. For this purpose, the chromatographic plates precoated with RP-18WF_254_ and Nano-Sil CNF_254_ were used, and the mobile phase consisted of methanol and water. The lipophilicity parameter (R_MO_) obtained by using Soczewiński-Wachtmeister’s equation on RP-18WF_254_ plates was 2.62 for indomethacin, 1.74 for ketoprofen and 2.71 for phenylbutazone. Meanwhile, the R_MO_ value estimated on Nano-Sil CNF_254_ plates for indomethacin, ketoprofen and phenylbutazone was 0.52, 2.31, as well as 1.50, respectively. Another author, Pehourq et al. [27], studied the lipophilicity expressed as log *k* of some NSAIDs, e.g., carprofen, flurbiprofen and ketoprofen, using the RP-HPLC technique, but mefenamic acid, nabumetone, indomethacin and phenylbutazone were not investigated. The following stationary phases, such as the ODS column prepacked with μBondapak C18 and an immobilized artificial membrane (IAM.PC.MG) column, were used in this study. The values of both the parameters log *k_w_*_ODS_ and log *k_w_*
_IAM_ were 3.85 and 1.81 for carprofen, 2.91 and 1.58 for flurbiprofen, and 2.83 and 1.02 for ketoprofen. Significant linear correlations (r > 0.94) between the chromatographic parameters (log*k_w_*
_IAM_) and the reference lipophilicity data (logP and logD_7.4_) were obtained.

In a work prepared by Czyrski [28], the lipophilicity among other NSAIDs like ketoprofen and flurbiprofen using RP-18F_254_ plates and the mobile phase composed of acetonitrile and water was studied. The R_m0_ values were determined for the compounds with a known logP and for ketoprofen (1.8491) and flurbiprofen (2.5076), respectively. Next, the lipophilicity parameters (logP) were calculated for the analyzed compounds using the prepared regression curve of type R_m0_ = *f*(logP). It was stated that flurbiprofen has the highest logP value, equal to 3.82, while ketoprofen has the lowest one, equal to 2.66.

As of now, there is no study in the scientific literature concerning the lipophilicity assessment of antihypertensive drugs, e.g., nilvadipine, felodipine, isradipine and lacidipine via the use of the TLC and calculation methods. Moreover, the values of the partition coefficient determined experimentally in the *n*-octanol-water system (logP_exp_) (available in the literature as well as online via databases) for drugs analyzed in this work belonging to antiparasitic and antihypertensive agents as well as NSAIDs are very diverse.

Therefore, the main purpose of this paper was the use of the RP-TLC technique and both Soczewiński-Wachtmeister’s and J. Ościk’s equations to investigate the lipophilicity of selected antiparasitic drugs (i.e., metronidazole, ornidazole, secnidazole, and tididazole), some antihypertensive drugs (i.e., nilvadipine, felodipine, isradipine, and lacidipine) and selected NSAIDs (i.e., mefenamic acid, nabumetone, phenylbutazone, carprofen, ketoprofen, flurbiprofen, and indomethacin)—see Appendix A.

## 2. Results and Discussion

In present work the lipophilicity of selected members belonging to three different groups of drugs, namely antiparasitic, antihypertensive and NSAIDs, were studied. For all examined compounds, the retention parameters R_M_ were determined by using the RP-TLC technique in various chromatographic systems. The obtained R_M_ values were used to determine two lipophilicity parameters marked as R_MWS_ and R_MWO_, respectively using Soczewiński-Wachtmeister’s and Ościk’s equations. On the basis of the results obtained, the lipophilic properties of the studied drugs were assessed.

The R_M_ values achieved under various chromatographic conditions were extrapolated to the zero content of the organic modifier (φ = 0) in the used mobile phase according to the Soczewiński-Wachtmeister’s equation. Next, the equations showing the linear relationship between the R_M_ values and organic modifier content in the used mobile phase were determined, and the R_MWS_ values for particular groups of examined compounds were estimated. Taking into account all cases, a high correlation coefficient *r* (above 0.95) was observed. All obtained lipophilicity parameters, including chromatographic parameters R_MWS_ and φ_0_, as well as R_MWO_, are presented in Table 1, Table 2 and Table 3.

Among the three groups of analyzed compounds, the highest values of the R_MWS_ lipophilicity parameter were obtained for antihypertensive drugs, and they ranged from 3.54 to 4.51 (in the acetone-water system) and from 3.13 to 5.01 (in the methanol-water system). For the examined NSAIDs, the R_MWS_ values range from 1.26 to 3.08 (in the acetone-water system) and from 1.47 to 3.12 (in the methanol-water system). However, the smallest R_MWS_ values were obtained for antiparasitic drugs, and they changed from 0.65 to 1.22 (in the acetone-water mobile phase) and from 0.91 to 1.46 (in the methanol-water system).

The R_MWO_ parameter for the three groups of analyzed compounds in the methanol-water and acetone-water systems on RP-18F_254_ plates was carried out in accordance with the methodology presented by Janicka [29,30]. The results of this parameter are placed in Table 1, Table 2 and Table 3.

The data listed in Table 3 indicate that there is a lack of R_MWO(a)_ values for a group of examined (NSAIDs) in the case of the acetone-water mobile phase, because this mixture was not suitable for testing the lipophilicity of these compounds. 

A further interpretation of the obtained results shows that among the three groups of analyzed compounds, the highest values of R_MWO_ were obtained for antihypertensive drugs in the methanol-water system as well as the acetone-water mobile phase. The R_MWO_ values are placed in the range of 5.57 to 6.63 (in the acetone-water system) and from 5.08 to 6.89 (for the methanol-water system). However, the smallest R_MWO_ values were observed for antiparasitic drugs. They are placed in the range of 0.94 to 1.42 (in the acetone-water system) and from 1.21 to 1.92 (in the methanol-water system).

The values of both the R_MWS_ and R_MWO_ parameters estimated for all analyzed compounds in the methanol-water and acetone-water systems indicate that the highest lipophilicity is shown by antihypertensive drugs, while the lowest lipophilicity belongs to the antiparasitic drugs. 

On the basis of the obtained results, it can be observed that the value of the chromatographic lipophilicity parameter R_MWS(a)_ = 0.65 for metronidazole is similar to R_MW(squalene)_ = 0.53 and R_MW(paraffin)_ = 0.60, which were given in the literature [23]. Meanwhile the value of the chromatographic lipophilicity parameter R_MWS(a)_ = 1.22 for ornidazole is comparable to the literature values, which were R_MW (undecane)_ = 0.98, R_MW (squalene)_ = 0.93 and R_MW (paraffin)_ = 1.12, respectively.

All parameters characterizing the lipophilic properties of tinidazole obtained in the present work by Soczewiński-Wachtmeister’s and Ościk’s methods have similar values to R_MW(undecane)_, R_MW(squalene)_ and R_MW(paraffin)_ [23], except for the R_MWO(m)_ value.

The lipophilicity parameters of metronidazole, ornidazole and tinidazole, estimated on the basis of Soczewiński-Wachtmeister’s and Ościk’s equations, using the RP-TLC technique (i.e., R_MWS_ and R_MWO_), are relatively higher than the results of the appropriate logk’ determined by Guerra et al. [25].

In the case of the studied antiparasitic drugs, the chromatographic parameters of these drugs and those available in the literature [23,25] differ from each other. The main reason for this fact can be, for example, the different adsorbing properties of applied stationary phases or eluents such as their elution strength or surface tension, which show an influence on the results of the R_M_ parameter. As was confirmed in previous studies, the pH value of distilled water used as a mobile phase component also has a large impact on the value of both the chromatographic lipophilicity parameters and the experimental partition coefficients [8,11,18,31].

The lipophilicity parameters determined on the basis of Soczewiński-Wachtmeister’s equation in the acetone-water and methanol-water systems (i.e., R_MWS(a),_ R_MWS(m)_) have values equal to 2.57 and 2.19 for indomethacin and 1.26 and 1.47 for ketoprofen, and they are close to the appropriate R_MO_ values determined on RP-18WF_254_ plates by Sârbu et al. [26].

All parameters describing the lipophilic properties of carprofen, flurbiprofen and ketoprofen estimated in this work via Soczewiński-Wachtmeister’s and Ościk’s methods have lower values in relation to the values of the appropriate log *k_w_*_ODS_ reported by Pehourq et al. [27], except for ketoprofen, for which the R_MWO(m)_ value equal to 3.76 is higher than the literature value of log *k_w_*_ODS_. However, the literature values of log*k_w_*
_IAM_ [27] have a lower value for carprofen, flurbiprofen and ketoprofen than the lipophilicity parameters of R_MWS_ and R_MWO_ determined in the current work via Soczewiński-Wachtmeister’s and Ościk’s methods [3,32].

The parameters characterizing the lipophilic properties of flurbiprofen and ketoprofen determined by Soczewiński-Wachtmeister’s and Ościk’s methods in this paper have lower values in relation to the values of logP reported by Czyrski [28]. The exception is ketoprofen, whose R_MWO(m)_ value is 3.76, which is higher than the literature values of logP.

The next stage of this paper was a comparison of all obtained R_MWS_ and R_MWO_ parameters calculated using Soczewiński-Wachtmeister’s and Ościk’s equations with theoretical and experimental logP values obtained in an *n*-octanol-water system. The theoretical partition coefficients, expressed as AlogPs, AC logP, milogP, AlogP, MlogP, XlogP2, and XlogP3, are shown in Table 1, Table 2 and Table 3. The experimental partition coefficients logP_exp_ available for all compounds except nilvadipine, lacidipine and carprofen, are presented in Table 4 [6,33,34,35,36,37,38,39,40,41,42,43,44].

Due to differences in the logP_exp_ values for metronidazole, ornidazole, tinidazole, felodipine, indomethacin, ketoprofen and flurbiprofen, the average values of logP_exp_ were calculated for these drugs.

Because the pH value of distilled water used to determine the partition coefficient in an *n*-octanol-water system may cause the differences in logP_exp_ values for the abovementioned compounds, the partition coefficient of the synthesized co-prodrug of flurbiprofen and methocarbamol was determined in the three mobile phases, namely: water, *n*-octanol-water (pH = 1.2) and *n*-octanol-phosphate buffer (pH = 7.4). Under these conditions the logP_o/w_ values equal to 5.13; 1.32 and 4.72 were observed [31].

Next, a cluster analysis by using the Statistica13.1 program was performed for three groups of analyzed drugs. This analysis precisely groups the most closely related parameters to be compared, like, for example, the described lipophilicity. For the purpose of the cluster analysis, all theoretical partition coefficients and chromatographic parameters, i.e., R_MWS_ and R_MWO_, except φ_0_, were used. Additionally, the calculated parameter φ_0_ cannot be compared with the logP values. It may be used for the purpose of a relative comparison of the lipophilic properties of the studied compounds only, and it is helpful in finding the order of lipophilicity degree of the examined group of compounds (from the most lipophilic to the least) [3]. The average values of logP_exp_ available for metronidazole, ornidazole, tinidazole, felodipine, indomethacin, ketoprofen, and flurbiprofen, and the values of logP_exp_ for other drugs (i.e., secnidazole, isradipine, mefenamic acid, nabumetone, and phenylbutazone), were also applied for this study. The results of the cluster analysis are presented as dendrograms in Figure 1, Figure 2 and Figure 3, respectively.

Figure 1 presents a comparison of the chromatographically determined lipophilicity parameters with the theoretical ones (Figure 1a) and logP_exp_ (Figure 1b) for the studied antiparasitic drugs. Figure 1 suggests that the parameters describing the lipophilicity of the studied antiparasitic drugs can be divided into two main subgroups. The first includes AlogPs, AClogP, XlogP3, milogP, AlogP, XlogP2 (Figure 1a) or logP_exp_, AlogPs, AC logP, XlogP3, milogP, AlogP, and XlogP2 (Figure 1b). The second group consists of MlogP and the chromatographic parameters of lipophilicity R_MW_, determined by Soczewiński-Wachtmeister’s and Ościk’s equations. Additionally, this diagrams shows that AClogP and AlogPs have the most similar values among the theoretical parameters. Meanwhile, the theoretical parameter MlogP shows the biggest similarity to the chromatographically determined lipophilicity parameter (R_MWS(a)_). The biggest similarity can also be observed between R_MWO(a)_ and R_MWS(m)_ in a group of chromatographically determined lipophilicity parameters. A significant difference in relation to other parameters is indicated by R_MWO(m)_, determined by the Ościk’s equation (Figure 1a,b) and logP_exp_, which separately forms a single subgroup (Figure 1b).

Next, Figure 2 represents a comparison of the chromatographically determined lipophilicity parameters with the theoretical ones (Figure 2a), as well as the logP_exp_ (Figure 2b) obtained for the antihypertensive drugs (except for nilvadipine and lacidipine), for which the logP_exp_ was not given in the literature.

Figure 2a shows that among the theoretical parameters, the biggest similarity indicates XlogP2 and XlogP3. Meanwhile, the theoretical parameter AlogPs is similar to the chromatographically determined lipophilicity parameter (R_MWS(m)_). Figure 2b suggests that the theoretical parameter XlogP2 indicates the biggest similarity to the logP_exp_ value. However, the AlogPs and AClogP values show the biggest connection to the chromatographically determined lipophilicity parameters (R_MWS(m)_ and R_MWS(a)_), respectively. The R_MWO(m)_ and R_MWO(a)_ determined by the Ościk’s equation form a separate subgroup (Figure 2a,b).

Next, Figure 3 shows the relation observed between the chromatographically determined lipophilicity parameters and the theoretical ones (Figure 3a) and the logP_exp_, respectively (Figure 3b) for the studied NSAIDs (except of carprofen, for which the logP_exp_ was not available).

Figure 3a,b demonstrate that the biggest similarity is indicated by the chromatographically determined lipophilicity parameters R_MWS(a)_ and R_MWS(m)_. The values of R_MWO(m)_ obtained by Ościk’s equation show the biggest connection to the theoretical parameters such as milogP and MlogP, due to the Euclidean distance. However, the logP_exp_ (Figure 3b) shows the biggest similarity to the XlogP3 and AlogPs, due to the Euclidean distances.

Taking into account a strong connection between the chromatographically determined lipophilicity parameters R_MWS_ and R_MWO_, respectively, as well as the theoretical and experimental parameters (logP) obtained for the three groups of analyzed drugs, it can be concluded that the R_MWS_ value may be a good alternative tool in describing the lipophilic nature of biologically active compounds having a high as well as low lipophilicity (e.g., antihypertensive drugs and antiparasitic drugs), respectively. On the other hand, R_MWO_ was found to be more suitable for the compounds with medium lipophilicity (i.e., NSAIDs).

The continuation of our lipophilicity study of fifteen biocompounds belonging to the three groups of drugs and the determination of another lipophilicity descriptor, namely φ_0_, in accordance with Equation (6), allowed us to obtain the two new lipophilicity parameters φ_0(a)_ and φ_0(m)_ for a proper used mobile phase, i.e., acetone-water (a) and methanol-water (m), respectively. What is important is that the calculation of this parameter was possible because a satisfactory linear relationship between both, i.e., the intercept and slope (R_MWS_ and S), in each obtained Soczewiński-Wachtmeister’s equation was achieved (r > 0.98). In order to estimate the utility of these parameters for the prediction of the theoretical as well as experimental parameter of the lipophilicity of a large group of investigated compounds, a correlation matrix between all previously described (thus chromatographic) parameters, as well as the computed and experimental partition coefficient, was done. The results of the correlation analysis indicate that among all φ_0_ values the best is the one obtained for acetone-water, which is to say φ_0(a)._ The most satisfactory linear correlations (r > 0.89) were obtained between the following partition coefficients: AClogP, XlogP3, logP_exp_ and the φ_0(a)_ values (see Equations (1)–(3):AClogP = -4.492(±0.951) + 10.312(±1.348) · φ_0(a)_(1)
n = 15, r = 0.905, s = 0.74, F = 58, *p* < 0.0001
XlogP3 = -4.937(±1.048) + 11.279(±1.484) · φ_0(a)_(2)
n = 15, r = 0.903, s = 0.82, F = 57, *p* < 0.0001
logP_exp_ = -4.491(±1.161) + 10.655(±1.685) · φ_0(a)_(3)
n = 12, r = 0.894, s = 0.89, F = 390, *p* < 0.0005

This fact confirms the potential utility of the additionally calculated chromatographic parameter φ_0(a)_ to determine the partition coefficient of all examined compounds. It can be very useful in the case of the lack of this value, like for example for the experimental logP of nilvadipine, lacidipine and carprofen. A test was performed on Equation (3) to determine the values of logP_exp_ for the three abovementioned compounds. The results of the predicted logP_exp_ were: 3.65 for nilvadipine, 3.96 for lacidipine and 3.54 for carprofen, respectively. It can be observed that the newly obtained logP_exp_ values are in good agreement with the other previously described theoretical logP of these compounds. They are placed in the similar range of logP values.

Summing up, the applied RP-TLC method and lipophilicity parameters denoted by R_MWS_ and R_MWO_ and calculated by using the retention parameter (R_M_) in accordance with Soczewiński-Wachtmeister’s and Ościk’s equations, respectively, may be the alternates to other lipophilicity descriptors (such as for example the logP determined by the classical *shake-flask* method) in describing the lipophilic character of bioactive compounds belonging to the following groups of drugs: antiparasitic, antihypertensive and non-steroidal anti-inflammatory drugs.

## 3. Materials and Methods

### 3.1. Chemicals and Standard Solutions

The standard solutions of all investigated compounds: metronidazole, ornidazole, secnidazole, tinidazole belonging to antiparasitic drugs; nilvadipine, felodipine, isradipine, lacidipine, used as antihypertensive agents; and a few non-steroidal anti-inflammatory drugs: mefenamic acid, indomethacin, nabumetone, phenylbutazone, carprofen, ketoprofen, and flurbiprofen, were supplied by Sigma-Aldrich (St. Louis, MO, USA). The solvents methanol and acetone, which have been used as mobile phase components, were from Merck (Darmstadt, Germany). The distillated water was from the Department of Analytical Chemistry (School of Pharmacy and the Division of Laboratory Medicine, Medical University of Silesia, Sosnowiec, Poland). Standard solutions of antiparasitic drugs and antihypertensive drugs at concentrations of 10 mg/mL each were prepared in methanol (Merck, Darmstadt, Germany). The solutions of indomethacin, nabumetone, phenylbutazone and ketoprofen at concentrations of 1 mg/mL each were prepared in acetone (Merck, Darmstadt, Germany). Standard solutions of carprofen and flurbiprofen at concentrations of 1 mg/mL each were prepared in ethanol (99.8%, POCh, Gliwice, Poland). A solution of mefenamic acid at a concentration of 1 mg/mL was prepared in a chloroform-methanol mixture in a volume ratio of 3:1. Chloroform was procured from POCh (Gliwice, Poland). All reagents had an analytical grade of purity.

### 3.2. RP-TLC

The chromatographic analysis was carried on RP-18F_254_ plates (Art.1.05559, E. Merck, Darmstadt, Germany). The solutions of the investigated compounds were spotted separately onto chromatographic plates using precise micropipettes in a quantity of 5 µL each. 

The chromatograms were developed using the mixtures of methanol-water and acetone-water in different volume compositions, as follows:

For antiparasitic drugs:-the content of methanol was gradually varied by 10% (*v*/*v*) from 20–100 (%, *v*/*v*),-the content of acetone was gradually varied by 10% (*v*/*v*) from 10–100 (%, *v*/*v*).

In the case of antihypertensive drugs, the content of methanol and acetone was gradually varied by 5% (*v*/*v*) from 60–100 (%, *v*/*v*).

For non-steroidal anti-inflammatory drugs:-the content of methanol was gradually varied by 5% (*v*/*v*) from 60–100 (%, *v*/*v*),-the content of acetone was gradually varied by10% (*v*/*v*) from 20–100 (%, *v*/*v*).

Fifty mL of used mobile phase was placed into a classical chromatographic chamber (Art. 022.5255, Camag, Muttenz, Switzerland). Next, the chamber was saturated with solvent vapor for 20 min. The chromatograms were developed at room temperature, e.g., 22 ± 1 °C. The development distance was 75 mm. The plates were dried at room temperature, e.g., 22 ± 1 °C. Each chromatogram was done in triplicate.

### 3.3. Densitometric Analysis

Densitometric scanning was done using a TLC Scanner 3 with WinCATS 1.4.2 software manufactured by Camag (Muttenz, Switzerland) in the reflectance/absorbance mode.

Densitometric scanning was performed at a respective absorption maximum for the analyzed drugs. The slit dimensions were 10.00 × 0.40 mm, Macro; the optimized optical system was light; the scanning speed was 20 mm/s; the data resolution was 100 μm/step; the measurement type was remission; the measurement mode was absorption; and the optical filter was second order. Each track was scanned three times, and a baseline correction (lowest slope) was used.

### 3.4. Calculations

#### 3.4.1. Chromatographic Parameters of Lipophilicity R_MWS_ and φ_0_

In order to determine the lipophilicity parameter based on Soczewiński-Wachtmeister′s procedure, the R_F_ values obtained under the applied chromatographic conditions were converted into R_M_ values according to the expression:(4)$$R_{M}=\log\left(1/R_{F}-1\right)$$
The linear relationship between R_M_ and the volume content of methanol and acetone in the mobile phase (φ) permits an extrapolation of the calculated R_M_ values to the zero concentration of methanol and acetone in accordance with Soczewiński-Wachtmeister′s Equation (5). The value of the intercept (R_MWS_) represents the lipophilicity parameter of the examined compound [2,3].
(5)$$R_{M}=R_{MWS}-S\varphi$$
where: R_M_ is the R_M_ value of the studied compound, R_MWS_ is the R_M_ value extrapolated to zero concentration of methanol and acetone (organic modifier) in the used mobile phase, i.e., methanol-water and acetone-water, respectively, S is the slope of the regression plot (see Appendix A), and φ is the volume fraction of methanol and acetone in the mobile phase.
In addition to this, on the basis of the said plot, i.e., Soczewiński-Wachtmeister′s equation, another chromatographic descriptor (φ_0_) has been calculated according to the following formula (Equation (6)):
(6)$$\varphi_{0}\;=\;\frac{R_{\mathrm{MWS}}}{S}$$
where: R_MWS_ is the chromatographic parameter obtained by using Soczewiński-Wachtmeister’s equation in accordance with Equation (5), and S is the regression slope.

#### 3.4.2. Chromatographic Parameter of Lipophilicity R_MWO_

The measurable lipophilicity value expressed as R_MWO_ was determined according to Ościk’s equation [29,30,32]:(7)$$G\left(x_{org}\right)=\frac{x_{org}\left(1-x_{org}\right)}{R_{M}-x_{org}R_{Morg}-\left(1-x_{org}\right)R_{MWO}}=ax_{org}+b$$
where: R_M_, R_Morg_, and R_MWO_ are the solute retention factors in the mixed mobile phase, pure organic solvent, and water, respectively; x_org_ is the molar fraction of the organic solvent in the mobile phase; a and b are constants in the linear correlation between G(x_org_) and x_org_ in the used mobile phase.

### 3.5. Determining the Theoretical and Experimental Partition Coefficients (logP)

The values of the theoretical partition coefficients, such as AlogPs, AClogP, AlogP, MlogP, XlogP2, and XlogP3, for the examined compounds, were obtained from the internet database VCCLAB [33]. The theoretical partition coefficient, milogP, for the examined compounds was from another internet database, namely Molinspiration Cheminformatics [45]. Therefore, different algorithms based on the chemical structure of all the tested compounds were applied for the prediction of the logP value [36,46]. 

The experimental logP_exp_ for the studied compounds were obtained from the scientific literature and internet databases [6,33,34,35,36,37,38,39,40,41,42,43,44], except for nilvadipine, lacidipine and carprofen, for which there is no data concerning their logP_exp_. 

### 3.6. Regression and Cluster Analysis (CA)

The regression and cluster analysis of the obtained results were performed with the use of the computer software STATISTICA 13.1.

## 4. Conclusions

The obtained results allow the following conclusions to be presented: 1.The RP-TLC method and the Soczewiński-Wachtmeister’s and Ościk’s equations are cost-effective and good tools for the assessment of the lipophilic properties of selected antiparasitic drugs such as metronidazole, ornidazole, secnidazole, and tinidazole, antihypertensive drugs like nilvadipine, felodipine, isradipine, and lacidipine, and non-steroidal anti-inflammatory drugs, i.e., mefenamic acid, nabumetone, phenylbutazone, carprofen, ketoprofen, flurbiprofen, and indomethacin.2.The mobile phase, consisting of acetone-water, is not suitable for the determination of the R_MWO_ parameter by using Ościk’s equation for the examined NSAIDs.3.The chromatographically determined lipophilicity parameters (R_MWS(a)_) show the biggest similarity to the theoretical parameter MlogP for antiparasitic drugs, which was calculated via the Moriguchi method.4.The two chromatographically obtained lipophilicity parameters (R_MWS(m)_ and R_MWS(a)_) show a strong connection with the theoretical parameters of lipophilicity, AlogPs and AC logP, in the case of antihypertensive drugs.5.The parameter R_MWO(m)_ determined by Ościk’s equation has the biggest similarity to the theoretical parameters milogP and MlogP for non-steroidal anti-inflammatory drugs due to the Euclidean distance.6.The chromatographic parameter φ_0(a)_ can be helpful for the prediction of partition coefficients, i.e., AClogP and XlogP3, as well as the logP_exp_ of examined compounds.

## Figures and Tables

**Figure 1 molecules-24-03187-f001:**
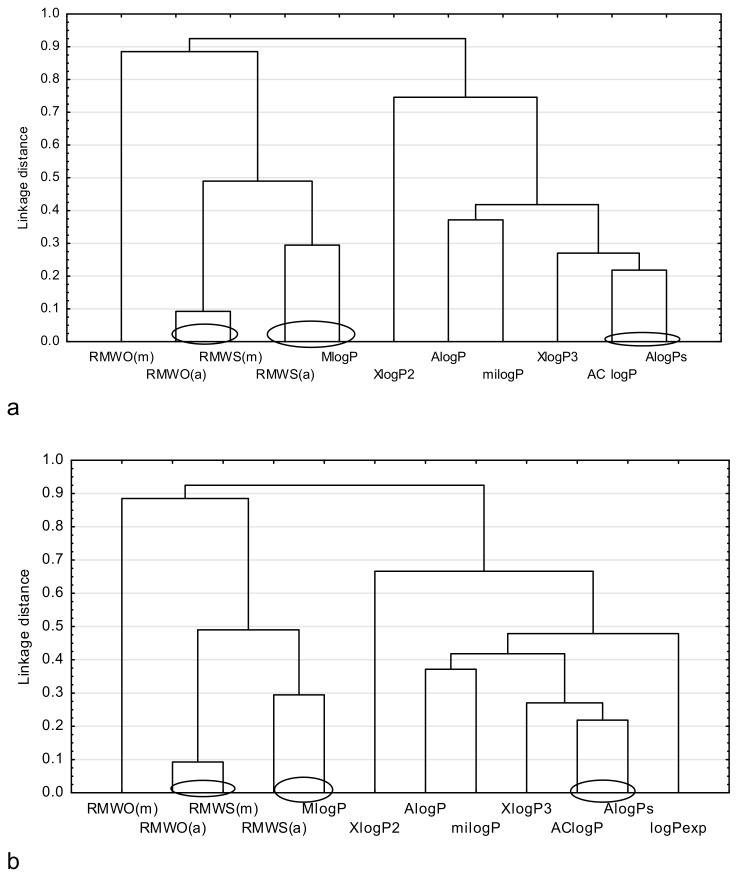
Similarity analysis for antiparasitic drugs, including: (**a**) comparison of the chromatographic and theoretical parameters of lipophilicity. (**b**) comparison of the chromatographic, theoretical and experimental parameters of lipophilicity.

**Figure 2 molecules-24-03187-f002:**
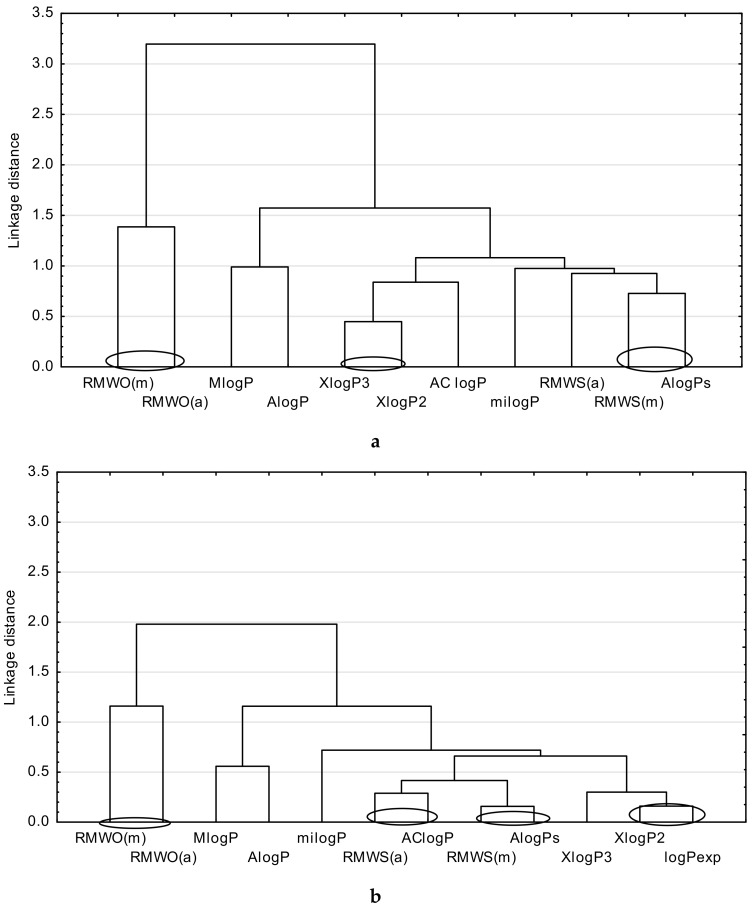
A similarity analysis for antihypertensive drugs, including: (**a**) comparison of the chromatographic and theoretical parameters of lipophilicity. (**b**) comparison of the chromatographic, theoretical and experimental parameters of lipophilicity.

**Figure 3 molecules-24-03187-f003:**
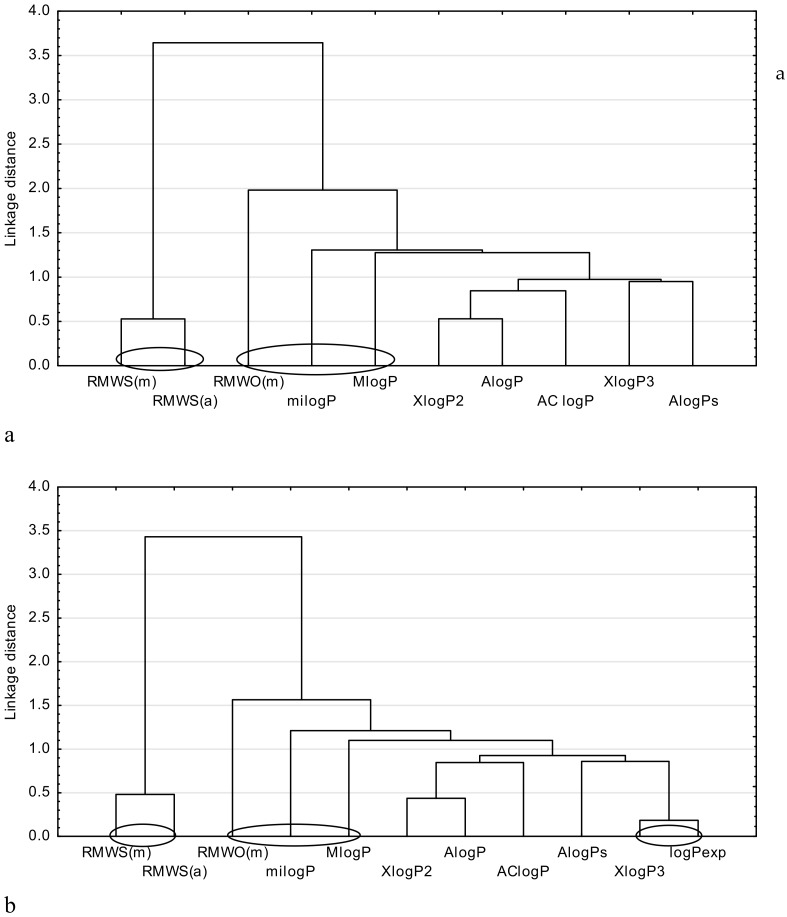
Similarity analysis for the examined NSAIDs, including: (**a**) comparison of the chromatographic and theoretical parameters of lipophilicity; (**b**) comparison of the chromatographic, theoretical and experimental parameters of lipophilicity

**Table 1 molecules-24-03187-t001:** Summary of the lipophilicity study of antiparasitic drugs by using the theoretical and RP-TLC methods.

Lipophilicity Parameters	Antiparasitic Drugs:			
	Metronidazole	Ornidazole	Secnidazole	Tinidazole
AlogPs	−0.15	0.37	0.25	−0.41
AC logP	−0.19	0.39	0.21	−0.20
milogP	−0.47	0.12	-0.10	−0.06
AlogP	−0.34	0.36	0.04	0.15
MlogP	0.44	1.13	0.80	1.00
XlogP2	−0.14	0.66	0.32	0.74
XlogP3	−0.02	0.60	0.22	−0.36
R_MWS(a)_ ^a^	0.65	1.22	0.95	0.89
R_MWS(m)_ ^b^	0.91	1.46	1.26	1.17
R_MWO(a)_ ^c^	0.94	1.42	1.21	1.11
R_MWO(m)_ ^d^	1.21	1.92	1.72	1.69
φ_0(m)_ ^e^	0.564	0.676	0.656	0.612
φ_0(a)_ ^f^	0.398	0.565	0.503	0.463

Where: ^a^ R_MWS(a)_—chromatographic lipophilicity parameter obtained experimentally on the basis of Soczewiński-Wachtmeister’s equation using acetone-water as the mobile phase on silica gel RP-18F_254_; ^b^ R_MWS(m)_—chromatographic lipophilicity parameter obtained experimentally on the basis of Soczewiński-Wachtmeister’s equation using methanol-water as the mobile phase on silica gel RP-18F_254_; ^c^ R_MWO(a)_—chromatographic parameter of lipophilicity obtained experimentally on the basis of Ościk’s equation using acetone-water as the mobile phase on silica gel RP-18F_254_; ^d^ R_MWO(m)_—chromatographic parameter of lipophilicity obtained experimentally on the basis of Ościk’s equation using methanol-water as the mobile phase on silica gel RP-18F_254_; ^e^ φ_0(a)_—chromatographic lipophilicity parameter calculated on the basis of the parameters of Soczewiński-Wachtmeister’s equation, i.e., R_MWS(a)_ and S, respectively using Equation (6); ^f^ φ_0(m)_—chromatographic lipophilicity parameter calculated on the basis of the parameters of Soczewiński-Wachtmeister’s equation, i.e., R_MWS(m)_ and S, respectively using Equation (6).

**Table 2 molecules-24-03187-t002:** Summary of the lipophilicity study of antihypertensive drugs by using the theoretical and RP-TLC methods.

Lipophilicity Parameters	Antihypertensive Drugs:			
	Nilvadipine	Felodipine	Isradipine	Lacidipine
AlogPs	2.97	4.36	3.00	5.18
AC logP	2.63	4.03	3.47	4.33
milogP	3.72	4.80	3.81	5.46
AlogP	2.09	3.55	2.17	3.82
MlogP	1.72	3.22	1.72	3.09
XlogP2	2.76	4.15	4.12	4.83
XlogP3	2.87	3.86	4.28	4.55
R_MWS(a)_ ^a^	4.07	3.75	3.54	4.51
R_MWS(m)_ ^b^	3.66	4.27	3.13	5.01
R_MWO(a)_ ^c^	6.28	5.57	5.67	6.63
R_MWO(m)_ ^d^	5.57	6.57	5.08	6.89
φ_0(m)_ ^e^	0.886	0.929	0.837	0.937
φ_0(a)_ ^f^	0.764	0.775	0.751	0.793

**Table 3 molecules-24-03187-t003:** Summary of the lipophilicity study of NSAIDs by using the theoretical and RP-TLC methods.

Lipophilicity Parameters	Non-Steroidal Anti-Inflammatory Drugs (NSAIDs):						
	Mefenamic Acid	Indomethacin	Nabumetone	Phenylbutazone	Carprofen	Ketoprofen	Flurbiprofen
AlogPs	4.58	4.25	3.41	2.81	4.09	3.29	3.57
AC logP	4.01	3.83	3.56	3.29	3.79	2.99	3.46
milogP	4.77	3.99	3.40	4.56	4.32	3.59	4.05
AlogP	3.96	4.21	2.80	3.95	4.09	3.34	3.66
MlogP	3.47	3.32	3.04	3.70	3.18	3.37	3.89
XlogP2	4.16	4.18	3.06	3.71	3.79	3.22	3.76
XlogP3	5.12	4.27	3.08	3.16	4.05	3.12	4.16
R_MWS(a)_ ^a^	2.61	2.57	3.08	1.72	2.56	1.26	1.87
R_MWS(m)_ ^b^	2.49	2.19	3.12	1.84	2.34	1.47	1.98
R_MWO(a)_ ^c^	-	-	-	-	-	-	-
R_MWO(m)_ ^d^	3.74	4.08	2.52	3.54	2.77	3.76	2.26
φ_0(m)_ ^e^	0.865	0.790	0.903	0.686	0.820	0.687	0.771
φ_0(a)_ ^f^	0.789	0.747	0.787	0.715	0.754	0.640	0.930

**Table 4 molecules-24-03187-t004:** The experimental partition coefficients (logP_exp_) of the examined compounds.

Drug	logP_exp_	$\log\bar{P}$
**Antiparasitic drugs:**		
Metronidazole	–0.22 [34] –0.10 [35]	
	–0.02 [36]	0.10 (± 0.44)
	0.75 [37]	
Ornidazole	0.23 [34]	
	0.59 [38]	0.41 (± 0.25)
Secnidazole	0.22 [33]	
Tinidazole	–0.35 [39]	
	0.70 [40]	0.18 (± 0.74)
**Antihypertensive drugs:**		
Nilvadipine	-	
Felodipine	3.86 [6]	
	4.46 [41]	4.16 (± 0.42)
Isradipine	4.28 [6]	
Lacidipine	-	
**Non-steroidal** **anti-inflammatory drugs** **NSAIDs:**		
Mefenamic acid	5.12 [39]	
Indomethacin	4.10 [42]	
	4.27 [36]	4.18 (± 0.12)
Nabumetone	3.08 [43]	
Phenylbutazone	3.16 [36]	
Carprofen	-	
Ketoprofen	3.12 [36]	
	3.14 [44]	3.13 (± 0.01)
Flurbiprofen	3.84 [42]	
	4.16 [36]	4.00 (± 0.23)

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
