# Peer review of "Comparison of the Utility of RP-TLC Technique and Different Computational Methods to Assess the Lipophilicity of Selected Antiparasitic, Antihypertensive, and Anti-inflammatory Drugs"

_molecules, 2019, doi:10.3390/molecules24173187_

Round 1

Reviewer 1 Report

Small corrections to make, first of all to shorten the introduction,  to be less verbose even in the presentation of the results.  Use abbreviations when already specified

Author Response

Comments and Suggestions for Authors

Small corrections to make, first of all to shorten the introduction,  to be less verbose even in the presentation of the results.  Use abbreviations when already specified

Responses of the Authors

Dear Reviewer ,

Authors would like to express the gratitude for the careful review of the paper.

According to valuable suggestions, the introduction part of this paper as well as presentation of results was modified (i.e. reduced) and appropriate abbreviations instead of long full name of drugs, chromatographic terms and etc. were introduced into  revised manuscript.

We hope that it will meet with your requirements.

Sincerely yours,

Alina Pyka-Pająk

Reviewer 2 Report

In the present study the authors report on the possibility of use of reversed-phase thin-layer chromatography (RP-TLC) to assess lipophilicity of four antiparasitic drugs, four antihypertensive drugs and seven non-steroidal autoinflammatory drugs. The authors use commercially available octadecyl-modified silica (RP-18) as a stationary phase and two mobile phase modifiers: acetone and methanol. Also, the authors use the extrapolated retention to the zero content of the mobile phase modifier (commonly expressed as RM0, or RMW value) as a lipophilicity measure of studied compounds. This parameter was estimated using two equations: Soczewiński-Wachtmeister’s and Ościk’s equations. Although, the physico-chemical interpretation of this parameter is the closest to the logarithm of partitioning coefficient between lipophilic and water environment, and therefore  the most similar to the traditionally used logP values, there are also, numerous chromatographic indices that can be used as a lipophilicity measures such as C0, b, mRM, scores of partial-least regression (PLS) and principal component analysis (PCA). In addition, the authors calculated seven computational logP values, and used experimentally determined logP values obtained from various literature sources for majority of studied compounds. The authors use hierarchical cluster analysis (HCA) to study similarities among the aforementioned lipophilicity parameters. The article is nicely written, and the references are properly cited.

However, despite the variety of compounds used, the study has some serious flaws.

First of all, extrapolated chromatographic values cannot be numerically compared with extrapolated values obtained from other chromatographic studies of the same compounds, simply because of the strong influence of variations in chromatographic conditions (temperature, shape and size of developing chamber, type of plates used  - even if they originate from the same producer a batch-to-batch differences can be significant, etc.).

Second of all, using only RMW parameter and discarding other chromatographic parameters that could better describe lipophilic character of studied compounds is a major drawback.

Third of all, HCA is only able to demonstrate similarities between studied lipophilicity measures but is not able to assess which of these methods is the best, or the worst, and which one is the most appropriate. Also, it is worth mentioning that in all studied classes of compounds HCA did not suggest strong similarity between any of the studied chromatographic lipophilicity measures and experimental logP. It seems that computational methods give more similar results to logPexp than any of the chromatographic experiments. Why is that?

The authors did not specify how the HCA was performed, i.e., what data transformation was used, what kind of distance and what sort of amalgamation rules were selected. There are many ways of doing so, and the results can differ significantly.

It is strongly advised to the authors to use more advanced methods based on non-parametric ranking to select the most suitable lipophilicity indices (the best and worst), to study a possible grouping among them, and to find the overall relationships (chromatograpgraphic vs. computational vs. shake-flask). Also, the authors should use more diverse set of chromatographic indices, not only the RMW

In addition to the acetone and methanol as mobile phase modifiers, the use of acetonitrile is recommended as it has the most prominent dipolar properties and the least proton-accepting, and proton-donating features (compared to methanol and acetone)

If the authors decide to carry out additional experiments, calculate additional chromatographic indices and perform more suitable statistical analysis such as non-parametric ranking, I would recommend this article for publication. However, in the present from, I do not find it suitable for publication, since it does not provide substantial contribution to the filed.

Author Response

Review 2

Dear Reviewer, the Authors would like to express the gratitude for the careful and inspiring review of the paper, in particular for emphasising that “the article is nicely written, and the references are properly cited.”

The manuscript has been accurately revised according to your valuable comments and suggestions. All changes made and responses to comments and suggestions are presented below. We hope that it will meet with your requirements.

Sincerely yours,

Alina Pyka-Pająk

Comment 1

In the present study the authors report on the possibility of use of reversed-phase thin-layer chromatography (RP-TLC) to assess lipophilicity of four antiparasitic drugs, four antihypertensive drugs and seven non-steroidal autoinflammatory drugs. The authors use commercially available octadecyl-modified silica (RP-18) as a stationary phase and two mobile phase modifiers: acetone and methanol. Also, the authors use the extrapolated retention to the zero content of the mobile phase modifier (commonly expressed as RM0, or RMW value) as a lipophilicity measure of studied compounds. This parameter was estimated using two equations: Soczewiński-Wachtmeister’s and Ościk’s equations. Although, the physico-chemical interpretation of this parameter is the closest to the logarithm of partitioning coefficient between lipophilic and water environment, and therefore  the most similar to the traditionally used logP values, there are also, numerous chromatographic indices that can be used as a lipophilicity measures such as C0, b, mRM, scores of partial-least regression (PLS) and principal component analysis (PCA). In addition, the authors calculated seven computational logP values, and used experimentally determined logP values obtained from various literature sources for majority of studied compounds. The authors use hierarchical cluster analysis (HCA) to study similarities among the aforementioned lipophilicity parameters. The article is nicely written, and the references are properly cited.

Responses of the Authors

Dear Reviewer, the main purpose of this work was comparison the utility of equations: Soczewiński–Wachtmeister’s as well as J. Ościk’s to estimate the lipophilicity of selected drugs of  antiparasitic, antihypertensive and non-steroidal anti-inflammatory drugs. For this reason we avoided the use of additional chromatographic parameters of lipophilicity and  partial-least regression (PLS) and principal component analysis (PCA). Moreover, application of  those additional parameters and mentioned statistical tools could increase the volume of manuscript. 

Comment 2

However, despite the variety of compounds used, the study has some serious flaws.

First of all, extrapolated chromatographic values cannot be numerically compared with extrapolated values obtained from other chromatographic studies of the same compounds, simply because of the strong influence of variations in chromatographic conditions (temperature, shape and size of developing chamber, type of plates used  - even if they originate from the same producer a batch-to-batch differences can be significant, etc.).

Responses of the Authors

Dear Reviewer, we would like to underline that our chromatographic studies are conducted in constant temperature (22±1oC). Our multiannual investigations (i.e. since 1985 year) indicate that shape and size of chromatographic chamber can affect the results of chromatographic analysis performed by using adsorption TLC only. Due to low pressure of mobile phases used in partition TLC (organic modifier+water), the shape and size of chromatographic chamber (development) does not affect the results of partition TLC, because the obtained results are reproducible and repectable. Thirty years ago, indeed there was lack of chromatographic plates dedicated for partition TLC. Thus, the researchers impregnated the silica gel 60 plates manually by means of paraffin oil or silicone oil, respectively. In this case, the obtained results weren’t precise. Introduction of chromatographic plates on market precoated with silica gel modified by the following groups: C18, C18W, C8, C2, CN, Diol, NH2 by Merck company has revolutionized the possibility of chromatographic studies by using TLC. Additionally, the use of densitometric analysis for the determination of accurate RF value directly after chromatographic plates development enables to obtain more reproducible so better quality results. Chromatographic plates by Merck, which have been used in these studies, namely RP-18F254 plates (Art.1.05559) are standardized and in combination with densitometric analysis ensure obtaining high quality and reproducible RF values with small error ±0.02. Our RM results were calculated accurately on the basis of RF values (Eq. 4). In addition to this, we would like to underline that the results of TLC analysis by using chromatographic plates from Merck company allow to obtain the precise and reproducible as well as robust on the changes of chromatographic conditions results. For example  the changes of chromatographic adsorbents (i.e. chromatographic plates 1.05554 and 1.05570 were shown in our previous papers: A.Pyka-Pająk, M. Dołowy, W. Parys, K. Bober, G. Janikowska, A Simple and cost-effective TLC-densitometric metod for the quantitative determination of acetylsalicylic acid and ascorbic acid in combined effervescent tablets, Molecules 2018, 23, 3115-3131; 2) Bocheńska, P.; Pyka, A. Determination of acetylsalicylic acid in pharmaceutical drugs by TLC with densitometric detection in UV. J. Liq. Chromatogr. Relat. Technol. 2012, 35, 1346–1363; 3) Bocheńska, P.; Pyka, A. Use of TLC for the quantitative determination of acetylsalicylic acid, caffeine, and ethoxybenzamide in combined tablets. J. Liq. Chromatogr. Relat. Technol. 2013, 36, 2405–2421; 4) Pyka, A.; Wiatr, E.; Kwiska, K.; Gurak, D. Validation thin layer chromatography for the determination of naproxen in tablets and comparison with a pharmacopeil method. J. Liq. Chromatogr. Relat. Technol. 2011, 34, 829–847. ).

Comment 3

Second of all, using only RMW parameter and discarding other chromatographic parameters that could better describe lipophilic character of studied compounds is a major drawback.

Responses of the Authors

Chromatographic parameter RMWS calculated according to Soczewiński-Wachtmeister’s equation was successfully used by our research group and others for  the estimation of lipophilic properties another classes of bioactive compounds. In current work, we have presented also the chromatographic parameter of lipophilicity calculated by means of  Ościk’s equation, namely RMWO. Additionally, accordance to  suggestion presented in Review Report No. 2, we have introduced  into the revised manuscript (see Tables 1-3) additional chromatographic parameter of lipophilicity, namely φ0. This parameter  was calculated as a ratio of RMWS and S values, where S is the regression slope in Soczewiński-Wachtmeister’s equation and RMWS - is the chromatographic parameter obtained by using this equation (see page 14). The values of S used to calculate this parameter are shown in supplementary materials. Chromatographic parameter of lipophilicity (φ0)  may be calculated when the examined group of drugs form a congeneric class, thus a linear relationships between the both i.e. RMWS values (intercept) and the slope of Eq. (5) S is observed. It was found in current work that the values of lipophilicity parameters RMWS and S obtained by using RP-TLC depends linearly on the slope of regression curve S (r>0.98).  Moreover, parameter φ0 can’t be compared with logP values. It may be used for the purpose of relative comparison of lipophilic properties of studied compounds and it is helpful in finding the order of lipophilicity degree of examined group of compounds (from the most lipophilic to less)  (Jóźwiak K., Szumiło H., Soczewiński E., “Lipophilicity, methods of determination and its role in biological effect of chemical substances. Wiad. Chem. 2001, 55, 1047 – 1074 (in Polish).

Comment 4

Third of all, HCA is only able to demonstrate similarities between studied lipophilicity measures but is not able to assess which of these methods is the best, or the worst, and which one is the most appropriate. Also, it is worth mentioning that in all studied classes of compounds HCA did not suggest strong similarity between any of the studied chromatographic lipophilicity measures and experimental logP. It seems that computational methods give more similar results to logPexp than any of the chromatographic experiments. Why is that?

Responses of the Authors

Dear Reviewer, in response to this comment, we would like to underline that the main purpose of our studies was to confirm the utility of partition thin-layer chromatography (RP-TLC) for the estimation of lipophilic properties of all examined compounds (i.e. different drugs) as not time-consuming and relatively economical method allowing to investigate all compounds (in our case 15 drugs) during the same experiment i.e. on the one chromatographic plates. Results of obtained chromatographic parameter of lipophilicity may be successfully applied for the estimation of drugs lipophilicity, especially when internet databases of theoretical logP values are not available.

Comment 5

The authors did not specify how the HCA was performed, i.e., what data transformation was used, what kind of distance and what sort of amalgamation rules were selected. There are many ways of doing so, and the results can differ significantly.

Responses of the Authors

 Dear Reviewer, for the purpose of presented HCA analysis, we have used the raw data (because the correlation matrix was not square), agglomeration method: single bond, measure distance: Euclidean distance.

Comment 6

It is strongly advised to the authors to use more advanced methods based on non-parametric ranking to select the most suitable lipophilicity indices (the best and worst), to study a possible grouping among them, and to find the overall relationships (chromatograpgraphic vs. computational vs. shake-flask). Also, the authors should use more diverse set of chromatographic indices, not only the RMW

Responses of the Authors

Dear Reviewer, in our paper we have used not one, but two chromatographic parameters of lipophilicity: RMWS  as well as RMWO calculated in accordance to equations:  Soczewiński-Wachtmeister’s  and J. Ościk’s. Moreover, based on suggestion presented in Review Report No. 2, we have introduced  into the revised manuscript (see Tables 1-3) another chromatographic parameter of lipophilicity, namely φ0. This parameter  was calculated as a ratio of RMWS and S values, where S is the regression slope in Soczewiński-Wachtmeister’s equation and RMWS – is the chromatographic parameter obtained by using this equation (see page 14). The values of S used to calculate this parameter are shown in supplementary materials. Chromatographic parameter of lipophilicity (φ0)  may be calculated when the examined group of drugs form a congeneric class, thus a linear relationship between the both i.e. RMWS values and the slope of Eq. (5) S is observed. It was found in current work that the values of lipophilicity parameters RMWS and S obtained by using RP-TLC depends linearly on the slope of regression curve S (r>0.98).  Moreover, parameter φ0 can’t be compared with logP values. It may be used for the purpose of relative comparison of lipophilic properties of studied compounds and it is helpful in finding the order of lipophilicity degree of examined group of compounds (from the most lipophilic to less).

Comment 7

In addition to the acetone and methanol as mobile phase modifiers, the use of acetonitrile is recommended as it has the most prominent dipolar properties and the least proton-accepting, and proton-donating features (compared to methanol and acetone)

Responses of the Authors

Dear Reviewer, in our lipophilicity study we have used also acetonitrile as organic modifier of applied mobile phase. However, this mobile phase consisted of acetonitrile+water was not suitable in most of the quality  compositions and affected the tailing of spots. Therefore, the obtained results were not reproducible. For this reason we have decided not to present these results.

Comment 8

If the authors decide to carry out additional experiments, calculate additional chromatographic indices and perform more suitable statistical analysis such as non-parametric ranking, I would recommend this article for publication. However, in the present from, I do not find it suitable for publication, since it does not provide substantial contribution to the filed.

Responses of the Authors

Dear Reviewer, according to suggestion, we have introduced  into the revised manuscript (see Tables 1-3) additional chromatographic parameter of lipophilicity, namely φ0. This parameter  was calculated as a ratio of RMWS and S values, where S is the regression slope in Soczewiński-Wachtmeister’s equation and RMWS – is the chromatographic parameter obtained by using this equation (see page 14). Chromatographic parameter of lipophilicity (φ0)  may be calculated when the examined group of drugs form a congeneric class, thus when the linear relationship between the both i.e. RMWS values and the slope of Eq. (5) S is observed. Moreover, parameter φ0 can’t be compared with logP values. It may be used for the purpose of relative comparison of lipophilic properties of studied compounds and it is helpful in finding the order of lipophilicity degree of examined group of compounds (from the most lipophilic to less). So, in present (i.e. revised work) we have used three different chromatographic parameters of lipophilicity obtained by using two mobile phases: methanol+water and acetone+water, respectively.

We have made also the additional calculation in form of correlation analysis for all drugs examined including all lipophilicity parameters and partition coefficients (theoretical and experimental).

Reviewer 3 Report

The authors describe how rp-TLC can be used to assess the polarity of certain drugs and compare this method with commonly used partition methods.

While the proposed method, reversed phase TLC, is indeed less labor intense and more applicable to directly compare polarity of compounds, it is by no means a new approach.

The authors don't explain the difference between (the more commonly used) Rf and Rn values (which are logarithmic forms of the Rf value)

The authors focus on the term "lipophilicity" rather than relating to "polarity" in general 

Only one theoretical parameter MlogPs seems to correlate with the Rn Values for antiparasitic drugs while the RlogPs relate better to the Rn values for antihypertensive drugs- (possible) explanation why this is the case, is missing.

Authors don't explain the difference of the individual theoretical parameters.

Author Response

Dear Reviewer ,

Authors would like to express the gratitude for the careful review of the paper. We have revised manuscript according to valuable suggestions. All changes and answers to questions are described below.

We hope that it will meet with your requirements.

Sincerely yours,

Alina Pyka-Pająk

Comment 1

The authors describe how rp-TLC can be used to assess the polarity of certain drugs and compare this method with commonly used partition methods.

While the proposed method, reversed phase TLC, is indeed less labor intense and more applicable to directly compare polarity of compounds, it is by no means a new approach.

Responses of the Authors

Dear Reviewer, there are many articles in the scientific literature that describing the application of the Soczewiński’s equation to assess the lipophilicity of various organic compounds, including drugs. However, the use of the Ościk’s equation to assess lipophilic properties of chemical compounds is  rarely described in the scientific literature.

Therefore, the present work is an novelty in the aspect of comparing the possibilities of using the Soczewiński’s and Ościk’s equations to study the lipophilicity of selected antiparasitic, antihypertensive and non-steroidal anti-inflammatory drugs.

Comment 2

The authors don't explain the difference between (the more commonly used) Rf and Rn values (which are logarithmic forms of the Rf value)

Responses of the Authors

The RF value is defined as the ratio between the distance from the origin to the spot center of the studied substance and the distance from the origin to the mobile phase. However, the definition of the RM value is given on page 14 of the manuscript. It is calculated  according to the  Eq. (4).                                                                   

Comment 3

The authors focus on the term "lipophilicity" rather than relating to "polarity" in general 

Responses of the Authors

We are focusing on the lipophilicity of compounds and not on their polarity due to the fact that lipophilicity:

1) is one of the most important physicochemical property of biologically active compounds, which has an effect on their biological activity,

2) enables predicting the absorption, distribution, metabolism and excretion profile (ADME profile), i.e. the behavior of the substance in the human body after its administration,

3) allows to determine the binding possibilities with the appropriate receptor,

4) the partition of the substance between the water and organic phases can be a partition model  in vivo.

As you known, the general condition for optimal absorption of the substance, e.g. from the gastrointestinal tract by passive diffusion after oral administration of the drug, is a moderate logP value (0-3 range).

 Comment 4

Only one theoretical parameter MlogPs seems to correlate with the Rn Values for antiparasitic drugs while the RlogPs relate better to the Rn values for antihypertensive drugs- (possible) explanation why this is the case, is missing.

Responses of the Authors

The drugs studied have different structures (see Supplemental Materials). However, theoretical partition coefficients are calculated using very different theoretical assumptions, also they contain experimental partition coefficients for selected substances, which are used to estimate the theoretical partition coefficient. Therefore, drugs divided at work in terms of their medicinal properties show different similarities to theoretical logP. The first group of compounds (antiparasitic drugs) are compounds belonging to 5-nitroimidazole derivatives and show the biggest similarity to MlogP. The second group of compounds are 3,5-pyridinedicarboxylic acid ester derivatives, which have similarities to AlogPs and AClogP. In contrast, NSAIDs do not have a common chemical structure; hence only RMWO(m) shows a little similarity to milogP on the dendrogram; while the Euclidean distances are much bigger for the other two parameters RMWS(m) and RMWS(a)   (they form a separate subgroup on the dendrogram).

Comment 5

Authors don't explain the difference of the individual theoretical parameters.

Responses of the Authors

 The  following calculation methods were used to calculate the values of the individual partition coefficients [1-3]:

AlogPs: the method was developed on the basis neural network ensemble analysis of more than 12000 organic compounds from the Physprop database using 75 E-state indices AClogP: the atom-additive method considering 369 atom-type-based contribution values milogP: the method for log P prediction developed at Molinspiration is based on group contributions, by fitting calculated log P with experimental log P for a training set more than 12000, mostly drug-like molecules AlogP: atomic contribution approach applying to neutral organic compounds containing among other N and halogen atoms MlogP: Moriguchi octanol-water partition coefficient is based on quantitative structure—log P relationships, by using topological indexes XlogP2: additive atom/group model which uses 90 basic atom types  XlogP3: knowledge-based approach based on the additive atom/group model which starts from the known log P value of a similarly reference compound

 Based on:

[1]       VCCLAB, “Virtual computational chemistry laboratory,” 2005, http://www.vcclab.org.

[2]       Katarzyna Bober , Ewa Bębenek, and Stanisław Boryczka, Application of TLC for Evaluation of the Lipophilicity of Newly Synthetized Esters: Betulin Derivatives, Journal of Analytical Methods in Chemistry Volume 2019, Article ID 1297659, 7 pages, https://doi.org/10.1155/2019/1297659

[3]       Alina Pyka, Magdalena Babuska, Magdalena Zachariasz, A comparison of theoretical  methods of calculation of partition coefficient for selected drugs, Acta Poloniae Pharmaceutica Drug Research, Vol. 63 No. 3 pp. 159-167, 2006

Reviewer 4 Report

The authors have described the use of reversed phase–thin layer chromatography method and the Soczewiński-Wachtmeister’s and Ościk’s equations for the assessment of lipophilic properties of several antiparasitic, antihypertensive drugs and non-steroidal anti-inflammatory drugs. Lipophilicity parameters of all examined compounds were obtained using Soczewiński-Wachtmeister’s and Ościk’s equations under various chromatographic systems and compared to different theoretical and experimental parameters.  The authors then determined the phase, parameters determined by this method that matches well with the parameters predicted by theory for different types of drugs. The manuscript is well written and provides details of a method that may become useful in future to determine lipophilicity in medicinal compounds. This manuscript can be published as is. 

Author Response

Comments and Suggestions for Authors

The authors have described the use of reversed phase–thin layer chromatography method and the Soczewiński-Wachtmeister’s and Ościk’s equations for the assessment of lipophilic properties of several antiparasitic, antihypertensive drugs and non-steroidal anti-inflammatory drugs. Lipophilicity parameters of all examined compounds were obtained using Soczewiński-Wachtmeister’s and Ościk’s equations under various chromatographic systems and compared to different theoretical and experimental parameters.  The authors then determined the phase, parameters determined by this method that matches well with the parameters predicted by theory for different types of drugs. The manuscript is well written and provides details of a method that may become useful in future to determine lipophilicity in medicinal compounds. This manuscript can be published as is. 

Responses of the Authors

Dear Reviewer,

Authors are very grateful for your positive estimation of this paper.

Sincerely yours,

On behalf of all Authors

Alina Pyka-Pająk

Round 2

Reviewer 2 Report

Dear authors,

Indeed, you have improved your work by adding one more lipophilicity measure. Still, I think that your work could be much better if other chromatographic lipophilicity measures were introduced. Nevertheless, it is your legitimate right to choose what you would like to study. Regarding the extrapolated chromatographic measures, they are still unreliable to be used as raw data. Otherwise the official OECD guidelines for chromatographic estimation of lipophilicity of chemical compounds would gladly recommend logkw to be used instead of logP. However, the guidelines are still recommending a series of standard compounds with known logP values to be always used for calibration. Nevertheless, the extrapolated chromatographic parameters are still frequently used as lipophilicity measures, and it is a reasonable aim to study which one of them is appropriate, and the most suitable. However, you simply cannot directly compare extrapolated measures obtained under different chromatographic conditions, using completely different techniques. This is completely wrong! For example, you cannot compare RMW values from the HPTLC experiments and the logkw from the HPLC experiment (a different technique, different properties of stationary phase, different mobile phase, etc.) Please see lines 193-196. Also, you made direct comparison with the values obtained by immobilized artificial membranes (completely different stationary phase and somewhat different separation mechanism). Please see lines 197-199. Moreover, you directly compared RMW values with logP (taken from the literature and obtained by the shake-flask method). Please see lines 200-203.  I simply cannot believe that someone so much experienced in chromatography and lipophilicity study can actually write all those statements!

Please consider removing these parts or completely rephrasing them!

And what does it mean if a certain value is higher or lower than the compared ones? Did you perform statistical test and confirm that such a difference is statistically significant? Comparing the numbers while neglecting the errors (or simple standard deviations) is pointless.

Also, it is still necessary to add detailed description of cluster analysis to the manuscript. It is not clear weather you standardized data, or used another scaling method, before you applied CA. Doing CA on the data with variables expressed in different scales can result in higher contribution of those variables with higher values. Also, it is not clear from the text what kind of distance between objects was used, e.g. Eucledian, Manhatten, City-block etc. Also, you have not clarified (in the manuscript) what amalgamation rule was used to build up the clusters, e.g. single linkage, complete linkage, centroid, weighted pair average linkage, or Ward’s method. The most frequently used distance and amalgamation rule are Eucledian distance and Ward’s method. This all should be explained in the experimental part as well.

In addition, why did you use a densitometric analysis? This is not clear from the experimental part. Usually, densitometry is performed in quantitative studies, while measurement of RF values can be done by a simple visual inspection of a chromatogram.